# Sodium Methoxide Catalysed One-Pot Glycidol Synthesis via Trans-Esterification between Glycerol and Dimethyl Carbonate

**Elrasheed Elhaj** [1,2]**, Huajun Wang** [1,3]**, Enaam A. Al-Harthi** [4,]*****, Waseem A. Wani** [5]**, Sahar Sallam** [6]**, Nasser Zouli** [7]
**and Mohd Imran** [7]

[1] Key Laboratory for Material Chemistry for Energy Conversion and Storage, Ministry of Education, School of Chemistry and Chemical Engineering, Huazhong University of Science and Technology, Wuhan 430074, China; wanghuajun@mail.hust.edu.cn (H.W.)

[2] Chemical Engineering Department, Faculty of Engineering and Technical Studies, University of El Imam El Mahdi, P.O. Box 209, Kusti 27711, Sudan

[3] Hubei Key Laboratory of Material Chemistry and Service Failure, School of Chemistry and Chemical Engineering, Huazhong University of Science and Technology, Wuhan 430074, China

[4] Department of Chemistry, College of Science, University of Jeddah, Jeddah 21959, Saudi Arabia

[5] Department of Chemistry, Government Degree College Tral, Srinagar 192123, India

[6] Department of Chemistry, College of Science, Jazan University, Jazan 45142, Saudi Arabia

[7] Department of Chemical Engineering, College of Engineering, Jazan University, Jazan 45142, Saudi Arabia

***** Correspondence: ealharthy@uj.edu.sa

**Abstract:** In this work we demonstrate one-pot glycidol synthesis, via trans-esterification between glycerol and dimethyl carbonate, by making use of commercially available sodium methoxide as a catalyst. An excellent glycerol conversion (99%) and remarkable glycidol yield (75%) was obtained using dimethyl carbonate/glycerol (molar ratio 2:1) in the presence of 3 wt% catalyst amount (with respect to glycerol weight) at 85 °C for a reaction time of 120 min. Sodium methoxide was recycled and reused twice with only a slight decrease in glycerol conversion. The water content of the glycerol reached 2.5 wt%; this did not reduce the glycerol conversion efficiency of the catalyst. A plausible mechanism for the trans-esterification involved in the preparation of glycidol was proposed.

**Keywords:** glycerol carbonate; one-pot glycidol synthesis; sodium methoxide-catalysed; stability of sodium methoxide catalyst; trans-esterification of glycerol

## 1. Introduction

Over the past few decades, there has been huge pressure on producers of biofuels, particularly biodiesel, due to the exceptionally high consumption of fossil fuels and major environmental degradation. Glycerol (GL) is formed as a co-product during biodiesel production and has thus been recently produced in huge quantities during the desired biodiesel production [1–3]. Therefore, the transformation of GL into a synthetically important product is one of the key approaches for the sustainable development of biodiesel industry [4–7]. Glycidol (GD) is one of the most important derivatives of GL. GD is used in the preparation and manufacture of polyurethanes, polyglycerols, glycidyl ethers, pharmaceuticals, surfactants, plastics, elastomers, paints, dye-levelling agents, etc. Furthermore, GD is also used as an important intermediate for the preparation of functional epoxides [8,9].

The industrial production of GD is ensured in two important ways. One of the procedures involves reacting 3-chloro-1,2-propanediol with bases, and the other one involves epoxidation of allyl alcohol using tungsten-based catalysts [10,11]. It is noteworthy that both the two procedures have quite a few drawbacks, such as high production costs, serious equipment corrosion, high liquid waste and chloride salt production, and consumption of starting materials derived from petrochemicals. The literature also describes GD synthesis by decarboxylation of glycerol carbonate (GC) [12–16]. In fact, GC has also been synthesized through the reaction of GL with dimethyl carbonate (DMC) or urea, and thus, this

route is basically a two-step synthetic procedure for the preparation of GD from GL and DMC [17,18]. In addition, the decarboxylation of GC is performed under harsh reaction conditions, viz., reduced pressure (~2.7 kPa) and high temperature range of 150–250 °C [19]. Since GD is highly reactive and readily polymerizes at high temperatures, this method may not be appropriate for the production of GD at industrial scale. Recently, GD has been prepared from GL and DMC by employing one-pot synthetic protocols [19–21]. In comparison to the above-mentioned methods for the preparation of GD, one-pot synthesis methods can be regarded as environmentally benign routes with numerous advantages such as mild operation conditions, high GD yields, and low toxicity of the raw materials. Up to now, certain catalysts such as tetramethylammonium hydroxide ionic liquid, tetraethylammonium amino acid ionic liquid, KF/sepiolite, and $NaAlO_2$, etc., have been successfully utilized for the synthesis of GD from GL and DMC using one-pot synthetic procedures [20]. Algoufi et al. investigated the one-pot preparation of GD by reacting GL with DMC using KF/sepiolite as catalyst. At optimum reaction conditions (4% catalyst weight loading, 2:1 DMC/GL molar ratio, 85 °C temperature, and 101.3 kPa pressure), 99% of GL conversion with 82.3% GD selectivity was reported [22]. Unfortunately, KF/sepiolite is a serious environmental pollutant since the fluoride ion ($F^-$) is easily lost; this can lead to the corrosion of equipment, along with serious environmental pollution. In addition, the ionic liquids are highly expensive and hence are not suitable for industrial applications. Moreover, $NaAlO_2$ easily undergoes hydrolysis, which greatly limits its industrial applications. Therefore, the search for developing new and effective catalysts for one-pot GD synthesis continues.

Sodium methoxide ($CH_3ONa$) is a base catalyst with high activity. It is successfully employed in the homogeneous catalytic production of biodiesel at an industrial scale [23–25]. We predicted that if sodium methoxide can be used for catalyzing one-pot GD synthesis by trans-esterification between GL and DMC, then both the industrial production of biodiesel and the conversion of GL can take place using the same catalyst. The overall process may greatly improve the industrial development of the one-pot synthetic protocols of GD. Until now, there have been no research reports on GD synthesis using sodium methoxide as catalyst. Therefore, in this paper, we demonstrate the one-pot GD synthesis by trans-esterification of GL with DMC using commercially available sodium methoxide as a base catalyst. All the parameters that affected GD yield were studied and a plausible reaction mechanism was also proposed.

## 2. Results and Discussion

### 2.1. Effect of Reaction Temperature

For GD synthesis, reaction temperature is an important factor affecting the rate of reaction. Generally, a high reaction temperature can reduce viscosity of the reaction mixture (especially for GL) and improve the diffusion process of reactants, thereby increasing the number of effective collisions among the reacting molecules; this consequently enhances the rate of reaction. However, GD polymerizes at high temperatures [22] so there must be an optimum temperature for trans-esterification between GL and DMC. Therefore, the temperature of the reaction was set in the temperature range of 75–90 °C at atmospheric pressure by keeping other variables constant. Figure 1 shows that GL conversion almost keeps constant (99%) upon increasing temperature from 75 °C to 90 °C. It indicated high activity of sodium methoxide for reacting GL with DMC. Upon increasing the reaction temperature from 75 °C to 85 °C, the yield of GD increased from 56–75% while the yield of GC decreased from 48–24%. With further increase in the reaction temperature to 90 °C, both the GD yield and GC yield were unaffected. Thus, 85 °C was selected as the optimum reaction temperature.

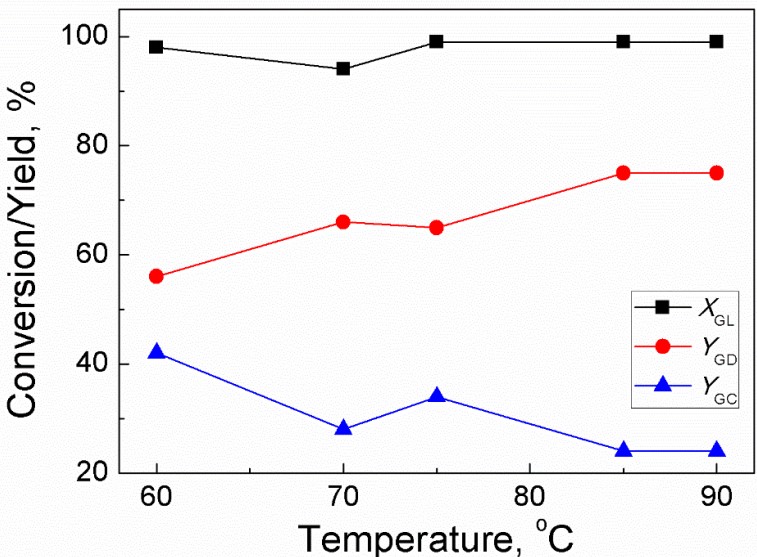

**Figure 1.** Effect of temperature on GL conversion and yields of GD and GC (Reaction conditions: DMC/GL molar ratio: 2:1; catalyst amount: 3 wt% (based on the weight of GL); time: 120 min).

## 2.2. Effect of Molar Ratio of DMC/GL

Since trans-esterification is a reversible reaction, the effect of the molar ratio of DMC/GL was studied in the range of 1:1 to 3:1 [26]. The results are shown in Figure 2. As the molar ratio of DMC/GL increased from 1:1 to 2:1, GL conversion and GD yield increased from 63% and 58% to 99% and 75%, respectively. However, further increase of molar ratio of DMC/GL could not significantly increase the conversion of GL. In addition, upon increasing the molar ratio to 3:1, the yield of GD decreased to 62% and an undesired product (glycerol dicarbonate) was detected. This meant that the excess of DMC shifted the equilibrium towards the formation of an undesired product. Thus, the optimum DMC/GL molar ratio was established to be 2:1.

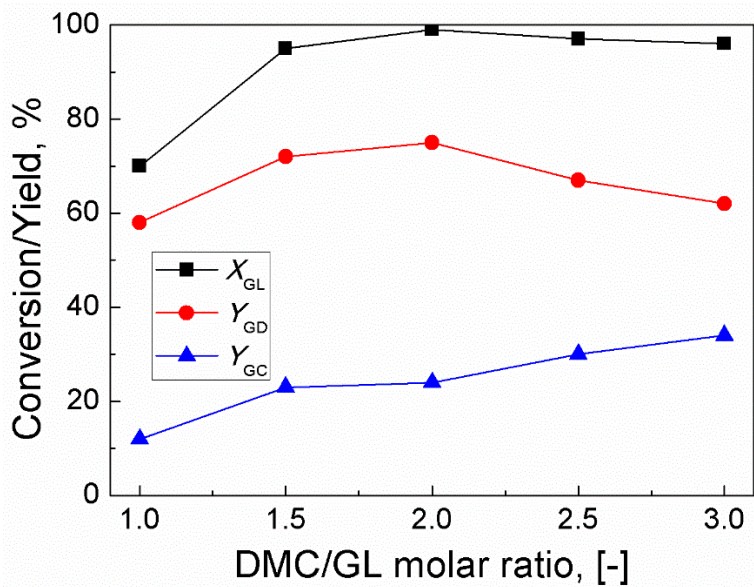

**Figure 2.** Effect of molar ratio of DMC/GL on GL conversion and yields of GD and GC (Reaction conditions: catalyst amount, 3 wt% (based on the weight of GL); time, 120 min; reaction temperature, 85 °C).

### 2.3. Effect of Catalyst Amount

The effect of catalyst amount on the reaction rate was studied by using an amount of catalyst in the range of 1–4 wt% of GL weight (Figure 3). The results showed that when the amount of catalyst increased from 1–3 wt%, the conversion of GL increased from 90–99% while the yield of GD increased from 38–75%. Further increase in the catalyst amount to 4 wt% led to a slim decrease in the yield of GD. In addition, when the catalyst amount increased from 1–4 wt%, the GC yield decreased from 52–24% firstly and then slightly increased to 29%. Therefore, the optimum catalyst amount was 3 wt% of the weight of GL. The data on effect of catalyst amount on GL conversion and yields of GD and GC has been provided in Table S1. Moreover, the calibration curves for Methanol, DMC, GC and GD can be found in Figure S2.

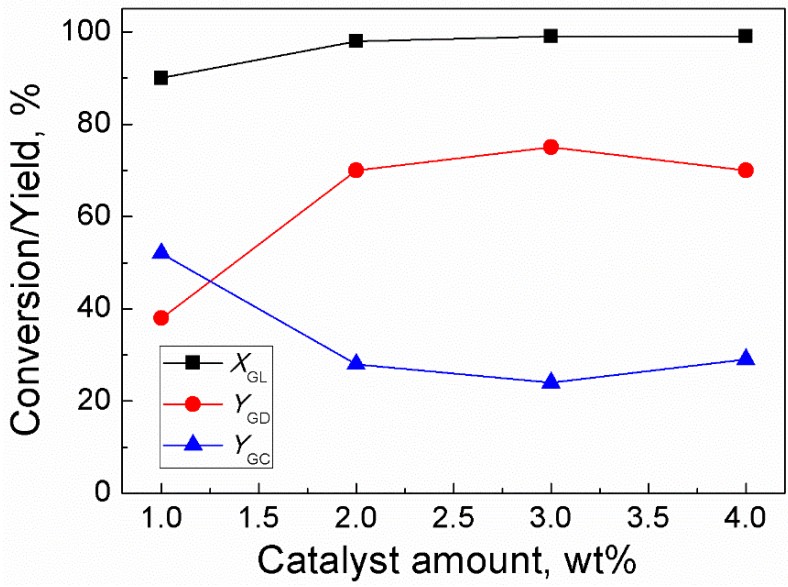

**Figure 3.** Effect of catalyst amount on GL conversion and yields of GD and GC (Reaction conditions: DMC/GL, 2:1; time, 120 min; temperature, 85 °C).

### 2.4. Effect of Reaction Time

For trans-esterification between GL and DMC, the effect of time was investigated under conditions in which molar ratio of DMC/GL was 2:1, catalyst amount was 3 wt% (of GL weight), and temperature was 85 °C (Figure 4). The reaction time was first increased to 40 min, and as a result, GL conversion and GD yield increased dramatically. As the reaction time was subsequently extended to 120 min, GL conversion and GD yield climbed gradually and steadily. The high GL concentration and high rate of reaction during the initial phases of the reaction were demonstrated by the rapid increase in GL conversion with increasing reaction time. However, over 40 min of reaction time, the reaction rate reduced because of diminished concentration of GL and as a result, the conversion of GL increased slowly. In addition, GC yield increased to a high value as the reaction time was increased to 20 min and then it gradually decreased to an almost stable value with a further increase in reaction time. This trend indicated that GD should have been generated by the decarboxylation of GC. Hence, the suitable reaction time for GD synthesis was found to be 90 min.

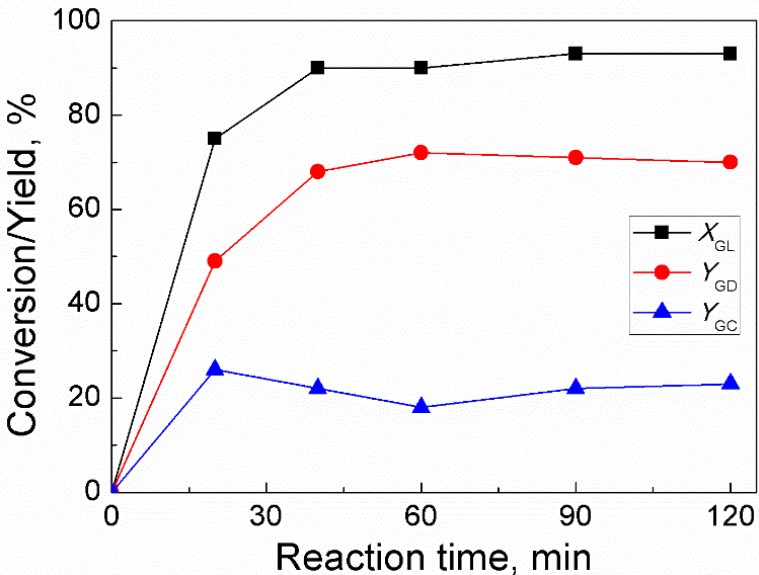

**Figure 4.** Effect of reaction time on GL conversion and yields of GD and GC (Reaction conditions: DMC/GL molar ratio, 2:1; catalyst amount, 3 wt% (based on the weight of GL); temperature, 85 °C).

### 2.5. Effect of Water Content in GL on Catalytic Performance of Sodium Methoxide

GL from the synthesis of biodiesel is used for GD synthesis. The so-obtained GL contains some water, hence its impact on the catalytic activity of sodium methoxide was investigated. The results are shown in Table 1. According to the results, the GL conversion and GD yield still reached 91 percent and 70 percent, respectively, even after increasing the water content in GL by up to 2.5 wt%. This indicated that the sodium methoxide catalyst had a stronger water-resistance performance than the CaO-based catalyst, for which GL conversion only reached about 55% when water content in GL was 2.0 wt% [27]. However, upon increasing water content in GL to over 5.0 wt%, the GL conversion and GD yield decreased rapidly; this meant that a large water content in GL is not suitable for the use of sodium methoxide as a catalyst.

**Table 1.** Effect of water content in GL on catalytic performance of sodium methoxide for the transesterification between GL and DMC.

| Water Content in GL (wt%) | $X_{GL}$ (%) | $Y_{GD}$ (%) | $Y_{GC}$ (%) |
| --- | --- | --- | --- |
| 0.0 | 99 | 75 | 22 |
| 2.5 | 91 | 70 | 21 |
| 5.0 | 76 | 60 | 16 |
| 7.5 | 69 | 57 | 12 |
| 11.0 | 61 | 41 | 20 |

Reaction conditions: DMC/GL molar ratio, 2:1; catalyst amount, 3 wt% (to GL weight); reaction temperature, 85 °C; reaction time, 120 min.

### 2.6. The Stability of Sodium Methoxide Catalyst

The stability of sodium methoxide catalyst was investigated under the following conditions: molar ratio of DMC/GL, 2:1; time, 120 min; temperature, 85 °C; and catalyst amount, 3 wt% (based on GL weight). In the first run, 2.0 g of GL, 3.91 g of DMC, and 0.06 g of fresh catalyst were charged. After reaction, the sample was collected from the reaction mixture and analyzed by gas chromatography. In the second run, 2.0 g of GL and 3.91 g of DMC were directly charged to the reaction mixture without separating the catalyst (which meant that in the second run, DMC/GL molar ratio had become 3:1). After reaction, a sample was also taken from the reaction mixture for analysis by gas chromatography and the results are depicted in Table 2. The chromatogram of the reaction mixture has been provided in Figure S1. Although the reaction in the second run faced

some opposing factors such as the dilution of reaction mixture and high concentration of product (produced in the first run), the GL conversion and GD yield still reached 95% and 73%, respectively. This shows that the catalyst has a high stability and can be reused. The GL conversion efficiency was lost by a very small amount (2%) for sodium methoxide catalyst in the second run; this is very small loss compared to the 11% loss in GL conversion efficiency for tetramethylammonium hydroxide used as catalyst for one-pot synthesis of GD by Gade et al. [28]. This indicates that sodium methoxide has a higher stability than tetramethylammonium hydroxide ionic liquid.

**Table 2.** The stability of sodium methoxide catalyst during the trans-esterification of GL with DMC.

| Recycle Number | $X_{GL}$ (%) | $Y_{GD}$ (%) | $Y_{GC}$ (%) |
|---|---|---|---|
| 1 (First Use) | 97 | 75 | 22 |
| 2 (Reuse) | 95 | 73 | 22 |

Reaction conditions: DMC/GL molar ratio, 2:1; catalyst amount, 3 wt% (to GL weight); reaction temperature, 85 °C; Reaction time, 120 min.

### 2.7. Comparison of Different Catalysts Used for GD Synthesis

One-pot GD synthesis from GL and DMC has been successfully achieved by using homogeneous, heterogeneous, and ionic-liquid based catalysts [19–23]. As shown in Table 3, sodium methoxide catalyst yielded the highest GL conversion (99%), which is the same as KF/sepiolite and is higher than the remaining three catalysts. Meanwhile, sodium methoxide yielded a moderate GD yield (75%), which is less than KF/sepiolite catalyst and is similar to the remaining catalysts. Thus, among the catalysts listed in Table 3, the activity of sodium methoxide is only slightly lower than KF/sepiolite. However, considering the potential corrosive effects of KF/sepiolite and its infancy as far as its development is concerned, sodium methoxide should perhaps be considered as more suitable for GD synthesis, as it has already been used at an industrial scale in biodiesel production.

**Table 3.** An analysis of catalytic efficiency of sodium methoxide with some other reported catalysts for the trans-esterification between GL and DMC.

| Catalyst | Reaction Type | Reaction Time (min) | $X_{GL}$ (%) | $Y_{GD}$ (%) | Reference |
|---|---|---|---|---|---|
| [a] NaAlO$_2$ | heterogeneous | 90 | 94.7 | 80.7 | [20] |
| [b] KF/sepiolite | heterogeneous | 90 | 99 | 82.3 | [22] |
| [c] Tetramethylammonium hydroxide | homogeneous | 90 | 95 | 78 | [28] |
| [d] Tetraethylammonium amino acid | homogeneous | 120 | 96 | 79 | [29] |
| [e] Sodium methoxide | homogeneous | 120 | 99 | 75 | Present work |

[a] DMC/GL molar ratio: 2:1; temperature: 80~90 °C; catalyst amount: 3 wt% based on GL weight. [b] DMC/GL molar ratio: 2:1; temperature: 83 °C; catalyst amount: 4 wt% based on GL weight. [c] DMC/GL molar ratio: 2:1; temperature: 80 °C; catalyst amount: 4 wt% based on GL weight. [d] DMC/GL molar ratio: 2:1; temperature: 130 °C; catalyst amount: 3 wt% based on GL weight. [e] DMC/GL molar ratio: 2:1; temperature: 85 °C; catalyst amount: 3 wt% based on GL weight.

### 2.8. Proposed Reaction Mechanism

Sodium methoxide is a typical homogeneous basic catalyst that has been used for catalyzing the trans-esterification of GL with DMC for GC and GD synthesis. A plausible mechanism for the trans-esterification can be proposed based on the reaction results and analysis of the published literature [6,23] (Scheme 1). The initial step involves the dissociation of sodium methoxide into methoxide anion ($CH_3O^-$) and $Na^+$ cation within the reaction mixture. In the second step, the primary O-H bond of GL gets activated by interaction with the methoxide anion that leads to the development of an activated state of GL along with formation of (O–H . . . O) hydrogen bond. In the third step the activated

GL attacks the carbonyl carbon of DMC to produce the intermediate methyl glyceryl carbonate and methanol. In the fourth step, methyl glyceryl carbonate further interacts with methoxide anion and ensures an intramolecular nucleophilic attack to generate GC and with the elimination of another methanol molecule. In the fifth step, a methoxide anion reacts with alkylene carbon of GC to form a ring-opening intermediate. In the final step of the reaction, decarboxylation of the ring-opened intermediate occurs via an intramolecular nucleophilic substitution reaction, forming GD and $CO_2$. In the meantime, the catalyst is also regenerated (see Scheme 1).

**Scheme 1.** The proposed plausible mechanism of trans-esterification between GL and DMC over sodium methoxide as catalyst. S1, S2, S3, S4, S5 and S6 represent first, second, third, fourth, fifth, and sixth steps of the reaction, respectively.

## 3. Materials and Methods

### 3.1. Chemicals

We used GC with a purity of over 90%, purchased from Tokyo Chemical Industrial Co., Ltd. Tokyo, Japan. GD with a purity of 97% was purchased from Shanghai SaEn Chemical Technology Co., Ltd., Shanghai, China. We purchased sodium methoxide (97%) from Aladdin Industrial Corporation in Shanghai, China. We obtained DMC and tetraethylene glycol (99%) from Tianjin Guangfu Fine Chemical Research Institute, Tianjin, China. We purchased GL and methanol, both 99%, from Sinopharm Chemical Reagent Co., Ltd., Beijing, China. We used 99% n-butanol that was procured from Shanghai Zhanyun Chemical Co., Ltd., Shanghai, China. The nitrogen was supplied by Sichuan Tianyi Science & Technology Co., Ltd., Sichuan, China. The nitrogen was 99.999 percent pure. All of the chemicals and reagents were used as they were received.

### 3.2. Reaction Procedure

Scheme 2 shows a schematic representation of the trans-esterification between GL and DMC that results in the production of GD that is catalyzed by sodium methoxide. A 50 mL 3-necked round bottom flask with a magnetic stirrer, a sampling device, a reflux condenser, and a thermocouple served as the experimental apparatus for the synthesis of GD. An oil

bath with a consistent temperature was used to warm the glass flask. GL and DMC were combined in exact ratios in each experiment run. Amounts of 2 g of GL and 3.91 g of DMC were added to the flask and heated to the desired temperature in a unique experimental run. The reaction was then started by adding a specified amount of sodium methoxide catalyst to the combination of GL and DMC (in a typical run, 0.06 g of sodium methoxide was used). In 120 min, the reaction took place. The magnetic stirrer's stirring speed was set to 600 rpm. The final reaction mixture was sampled for examination once the reaction was finished.

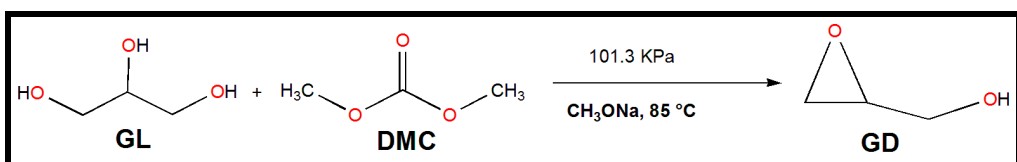

**Scheme 2.** One-pot GD synthesis from GL and DMC using commercially available powdered sodium methoxide as a catalyst.

*3.3. Product Analysis*

All the components were analyzed using a gas chromatograph (Fuli 9790-II) equipped with a flame ionization detector (FID) and capillary column KB-WAX (30 m long, 0.25 mm i.d.). *n*-Butanol (internal standard) was used for determining DMC, methanol, and GD. Tetraethylene glycol was employed for determination of GL and GC. Nitrogen was used as a carrier. The pressure was adjusted to 0.3 MPa and the flow rate was set at 30 mL/min. The injector and detector were designed to operate at temperatures of 250 °C and 270 °C, respectively. The column temperature was held at 70 °C for 2 min before ramping up to 250 °C for 15 min. Each component of the reaction mixture has a respectable peak separation. GL conversion ($X_{GL}$), GD yield ($Y_{GD}$), and GC yield ($Y_{GC}$) were calculated from the following equations:

$$X_{GL} = \frac{m^0_{GL} - m^t \cdot c^t_{GL}}{m^0_{GL}} \times 100\% \tag{1}$$

$$Y_{GD} = \frac{m^t \cdot c^t_{GD}/M_{GD}}{m^0_{GL}/M_{GL}} \times 100\% \tag{2}$$

$$Y_{GC} = \frac{m^t \cdot c^t_{GC}/M_{GC}}{m^0_{GL}/M_{GL}} \times 100\% \tag{3}$$

where $m^0_{GL}$ is the initial mass (in grams) of GL and $m^t$ is the total mass (in grams) of the residual reaction mixture after reaction. $c^t_{GL}$, $c^t_{GD}$, and $c^t_{GC}$ is the GL concentration (wt%), GD concentration (wt%), and GC concentration (wt%) in the residual reaction mixture, respectively, while $M_{GL}$, $M_{GD}$, and $M_{GC}$ are the molar masses in grams/mol of GL, GD and GC, respectively.

**4. Conclusions**

Sodium methoxide is used at industrial scale in the production of biodiesel. The worth of the catalyst is further amplified in this experiment as it efficiently catalyzes the one-pot GD synthesis via the trans-esterification between GL and DMC. The optimum reaction conditions for the trans-esterification between GL and DMC are (molar ratio of DMC/GL: 2:1, 3 wt% catalyst amount, temperature: 85 °C, and time: 120 min). Sodium methoxide efficiently catalyzed the reaction and ensured a GL conversion of 99% and GD yield of 75%. In addition, the catalyst was reused twice with only a slightly decrease in GL conversion and GD yield. Furthermore, a 2.5 wt% water content in GL did not show any noticeable effect on the efficiency of GL conversion of sodium methoxide. The catalysis of the reaction is initiated by dipole-dipole interaction (hydrogen bonding) of the catalyst with GL followed by a series of nucleophilic, elimination, and ring-opening

events of the intermediates. Overall, the reported sodium methoxide catalyzed one-pot trans-esterification between GL and DMC for GD synthesis is a facile approach for the production of GD and may be a quite useful procedure at industrial scale.

**Supplementary Materials:** The following supporting information can be downloaded at https://www.mdpi.com/article/10.3390/catal13050809/s1, Figure S1: The gas chromatogram of the reaction mixture for synthesis of GD from GL and DMC using $CH_3ONa$ as catalyst: (1) methanol; (2) DMC; (3) n-Butanol; (4) GD; (5) GL; (6) tetraethylene glycol; (7) GC.; Figure S2: Calibration Curve (a) Methanol, (b) DMC, (c) GC and (d) GD; Table S1: Data for the effect of catalyst amount on the GL conversion and yields of GD and GC.

**Author Contributions:** Conceptualization, E.E. and H.W.; methodology, E.E.; software, W.A.W.; validation, E.E., H.W. and E.A.A.-H.; formal analysis, S.S., N.Z. and M.I.; investigation, E.E., E.A.A.-H. and H.W.; data curation, E.A.A.-H., S.S. and N.Z.; writing—original draft preparation, E.E.; writing—review and editing, E.E., H.W., W.A.W. and M.I.; visualization, W.A.W.; supervision, H.W.; project administration, H.W. and E.A.A.-H.; funding acquisition, E.A.A.-H. All authors have read and agreed to the published version of the manuscript.

**Funding:** The authors extend their appreciation to the Deputyship for Research & Innovation, Ministry of Education in Saudi Arabia for funding this research work through project number MoE-IF-UJ-R-22- 04101095-1.

**Data Availability Statement:** The authors state that the data pertaining to the manuscript will be made available upon request.

**Acknowledgments:** The authors extend their appreciation to Analytical and Testing Center, Huazhong University of Science and Technology, Wuhan, China and University of El Imam El Mahdi, Kosti, Sudan.

**Conflicts of Interest:** There are no conflict of interest to declare.

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
