# Peer review of "Sodium Methoxide Catalysed One-Pot Glycidol Synthesis via Trans-Esterification between Glycerol and Dimethyl Carbonate"

_catalysts, doi:10.3390/catal13050809_

Round 1

Author Response

Thank you for helpful comments and valuable suggestions on our manuscript. We have carefully revised the manuscript and addressed all the comments provided by editors and reviewers and replied point-by-point.  

Reviewer #1

Comment 1: Arrange your Keywords in alphabetical order (lines 23-29).

Response: Thank you for your comment. We have arranged keywords in alphabetical order as follows, Glycerol carbonate; One-pot glycidol synthesis; Sodium methoxide-catalysed; Stability of sodium methoxide catalyst; Trans-esterification of glycerol.

Comment 2:   You are supposed to use “from” instead of “form” which was used in the sentence in line 52 (… two-step synthetic procedure for the preparation of GD form GL and DMC).

Response: Thank you for your kind notice. It has been corrected in the revised manuscript.

Comment 3:  Please could you give any scientific explanation for the slight decrease in the GD yield and slight increase in the GC yield after a further increase in the catalyst amount to 4 wt%?

Response: Thank you so much for your valuable comment and query.  The GD yield decreased while studying the stability of catalyst, because we estimated that the DMC/GL molar ratio became close to 3:1, but actually it is less than 3:1, resulting in GDC formed as by-product. In addition, on increasing the amount of catalyst, both the conversion of GL and the yield of GD were affected and all GL was converted, it might increase the conversion of GC into GD by decarboxylation.

Comment 4:  Why was the reusability test not done for the number of cycles of 4 just as it was done in the article by Y.T. Algoufi et al. /Applied Catalysis A: General 487 (2014) 181–188? Having the reusability test done for the number of cycles of 4 will help to show clearly its level of stability.

Response: Thank are due to the learned reviewer for this comment. The reuse of the catalyst was done by adding new reactant without the separation of catalyst, which was difficult, so the reaction mixture was diluted that led to increase of the byproduct, GDC. GDC formation can be avoided at conversions below 100%, In addition, there are additional data for the effect of catalyst amount on the GL conversion and yields of GD and GC, which indicated that most glycerol was converted to GC and GD.

Catalyst amount of sodium methoxide (54.02g/mol)

Glycerol (92.08g/mol), =2/92.08 = 0.02172024

Data for the effect of catalyst amount on GL conversion and yields of GD and GC

Catalyst %w

Cat. g

Cat. mol

Mol%

Conversion % of GL

Yield of GD %

Yield of GC %

1

0.02

0.00037

1.705

90

38

52

2

0.04

0.00074

3.410

98

70

28

3

0.06

0.00111

5.115

99

75

24

4

0.08

0.00148

6.820

99

70

29

5

0.10

0.00185

8.525

99

84

15

Comment 5: Please correct the GD yield achieved with KF/sepiolite from 81.5% to 82.3% in Table 3.

Response: Thank you for your notice. It has been corrected as 82.3% in the revised manuscript.

Comment 6:   I recommend that the use of the English language is checked.

Response: We have carefully revised the manuscript and avoided any grammar or syntax errors. In addition, we have taken help from English native for language improvement. We believe that the language in the revised manuscript is acceptable.

Reviewer 2 Report

Line 26 remove word “remarkably”

Line 33, 34 the sentence should be rewritten

Line 40 remove word “profoundly”

Line 87-97 Experimental part should be rewritten and changes sentences” from was bought” to directly to “was used”.

Scheme 2. is displaced

Authors should add a chromatogram of the reaction mixture and calibration curves (at least in supplement)

Author Response

Thank you for helpful comments and valuable suggestions on our manuscript. We have carefully revised the manuscript and addressed all the comments provided by editors and reviewers and replied point-by-point.  

Reviewer #2

Comment 1: Line 26 remove word “remarkably”

Response: Thanks for your comment. The word “remarkably” was removed in revised manuscript.

Comment 2: Line 33, 34 the sentence should be rewritten

Response: Thanks so much for your valuable comments. We have rewritten the sentence as follows-Over the past few decades, there has been huge pressure on producers of biofuels, particularly biodiesel, due to the exceptionally high consumption of fossil fuels and major environmental degradation.

Comment 3: Line 40 remove word “profoundly”

Response: Thank you for your comment. The word has been removed.

Comment 4: Line 87-97 Experimental part should be rewritten and changes sentences” from was bought” to directly to “was used”.

Response: Thank you for your suggestion. We accepted the suggestion made by the reviewer. In revised manuscript, Experimental part has been rewritten and changed sentences according to reviewer’s comments. In the revised manuscript, according to new format requested by editorial team, the revised text is now at lines 226-235. The revised text is as follows:

We used GC with a purity of over 90%, purchased from Tokyo Chemical Industrial Co., Ltd. Tokyo, Japan. GD with a purity of 97% was purchased from Shanghai SaEn Chemical Technology Co., Ltd. Shanghai, China. We purchased sodium methoxide (97%) from Aladdin Industrial Corporation in Shanghai, China. We obtained DMC and tetraethylene glycol (99%) from Tianjin Guangfu Fine Chemical Research Institute, China.  We purchased GL and methanol, both 99%, from Sinopharm Chemical Reagent Co., Ltd. China. We used 99% n-butanol that was procured from Shanghai Zhanyun Chemical Co., Ltd. China. Nitrogen was supplied by Sichuan Tianyi Science & Technology Co., Ltd. Sichuan, China. Nitrogen was 99.999 percent pure. All of the chemicals and reagents were used as they were received.

Comment 5: Scheme 2. is displaced

Response: All schemes and all images are now in the correct format and tightly wrapped.

Comment 6: Authors should add a chromatogram of the reaction mixture and calibration curves (at least in supplement).

Response: We are thankful to the reviewer for this important suggestion. We have added a gas chromatogram of the reaction mixture and calibration curves in supplementary information.

Reviewer 3 Report

This manuscript describes glycidol synthesis via Trans-esterification between glycerol and dimethyl carbonate. The data are poorly presented, and the schemes are poorly drawn. And with NaOMe and water co-existed at such a high temperature, I would consider the real catalyst is NaOH, and MeOH probably already generated and evaporated. Then the reaction is simply a NaOH mediated trans-esterification, which is well-known by high school students. All the other studies like reusability studies etc look like pure exercises in style and does not make the current manuscript looks better.

Author Response

Thank you for helpful comments and valuable suggestions on our manuscript. We have carefully revised the manuscript and addressed all the comments provided by editors and reviewers and replied point-by-point.  

Reviewer #3

Reviewer #3, Comments and Suggestions for Authors

This manuscript describes glycidol synthesis via trans-esterification between glycerol and dimethyl carbonate. The data are poorly presented, and the schemes are poorly drawn. And with NaOMe and water co-existed at such a high temperature, I would consider the real catalyst is NaOH, and MeOH probably already generated and evaporated. Then the reaction is simply a NaOH mediated trans-esterification, which is well-known by high school students. All the other studies like reusability studies etc. look like pure exercises in style and does not make the current manuscript looks better.

Response: Thanks for your comments, firstly, Methoxide (CH3O−) is the conjugate base of methanol. Methanol is very weak acid (e.g. its dissociation constant is very small, Pka=16), so its conjugated base is very strong and a good nucleophile, however, with a basic catalyst such as hydroxide ion, the first step in enolization is removal of a proton from the alpha position to give the enolate anion 1. Normally, C-H bonds are highly resistant to attack by basic reagents, but removal of a proton alpha to a carbonyl group results in the formation of a considerably stabilized anion with a substantial proportion of the negative charge on oxygen, as represented by the valence-bond structure below.

deprotonation of the OH group of the pending ­CH2OH moiety may occur, with epoxidation implying the newly formed CH2O­ anion, followed by CO2 elimination on the other end of the molecule. So, we do not think that NaOH is the catalyst in this reaction however, the boiling point of NaOMe is 95oC and the reaction was performed in the range of temperature between 60~90 oC at atmospheric pressure.

The scheme 1 (previously scheme 2) is redrawn and presented in the revised manuscript.

Round 2

Reviewer 3 Report

The authors could not address the previous questions and concerns previously proposed.

First of all, everybody knows methoxide (CH3O−) is the conjugate base of methanol and the following stuff the authors provided. There’s no basic logic in the authors’ explanation, the authors should run their reaction using the same molar amount of NaOH to prove methoxide anion does have a role here. Since the boling point of methanol is 65 degC, the authors said that the reaction was performed in the range of temperature between 60~90 degC at atmospheric pressure. It is totally reasonable that NaOMe reacts with water, giving NaOH and also MeOH as a gas, which is then evaporated. The mechanism should be similar like the previous publication: (https://downloads.hindawi.com/journals/ijps/2021/9300442.pdf), And as I said there’s nothing new here.

Moreover, the authors don’t know how to do arrow pushing for an organic reaction. For example, in Mechanism scheme, in S3, the arrow should start from the glycerol OH not the methoxide. For S4, the arrow should not point the bond but the carbon center. S5 and S6 looks even more weird. If you do arrow pushing, you should do it right. And the authors should not use such gigantic and messy structures in your equations, you can use either ACS style or other style that looks good.

The transformation is not new at all, as long as you have a base, you should somewhat catalyze this reaction, please check previous publications like, 1, Green Chem 10.1039/c2gc35992h 2, Catal. Sci. Technol., 2013, 3, 3242.

I would be surprised if this kind of work can be accepted by such a decent journal.

Author Response

Response to Decision Letter

Manuscript (Catalysts-2301571)

Thank you very much for the helpful comments and valuable suggestions for the improvement of our manuscript. We have carefully revised the manuscript and addressed all the comments provided by the reviewer. The replies to the comments of the reviewer have been made point-by-point.  

Reviewer #3

Comment 1: The authors could not address the previous questions and concerns previously proposed. First of all, everybody knows methoxide (CH3O-) is the conjugate base of methanol and the following stuff the authors provided. There’s no basic logic in the authors’ explanation, the authors should run their reaction using the same molar amount of NaOH to prove methoxide anion does have a role here. Since the boiling point of methanol is 65 C, the authors said that the reaction was performed in the range of temperature between 60~90 C at atmospheric pressure. It is totally reasonable that NaOMe reacts with water, giving NaOH and also MeOH as a gas, which is then evaporated. The mechanism should be similar like the previous publication: (https://downloads.hindawi.com/journals/ijps/2021/9300442.pdf), And as I said there’s nothing new here.

Response: Thanks are due to the learned reviewer for raising this point. The knowledgeable reviewer has correctly stated that methoxide anion is the conjugate base of methanol. The learned reviewer has further rightly stated that NaOMe reacts with water, giving NaOH and also MeOH as a gas, which is then evaporated. It is important to note that sodium methoxide is a stronger base than sodium hydroxide. Also, methoxide anion is a better nucleophile than hydroxide. The trans-esterification reaction reported in this work has been carried out in absence of water. However, the reaction has been studied with different water contents in glycerol. A keen look into the discussion in the manuscript (Please see Table 1) shows that increase in the water content in GL has had a pronounced effect on the decrease in the glycerol conversion, glycidol yield and glycerol carbonate yield, which might be attributed to the reaction of NaOMe with water and the formation of NaOH, and the latter’s participation in the catalysis of the reaction (as pointed out by the learned reviewer). As already highlighted above, NaOH is a weaker base and nucleophile than sodium methoxide, the reaction catalyzed by NaOH has therefore, led to decreased efficiency of the reaction. However, the participation of NaOH in the catalysis of the reaction takes place for an insignificant time and proportion (as the water content in glycerol is truly very small), and hence it is not appropriate to say that NaOH is the catalyst of the reaction.

In one of our previous papers (Chemical Engineering Journal, 2022, 442, 136196), KOH was used to catalyze the reaction of glycerol with dimethyl carbonate to generate glycidol, whose activity was lower than that of sodium methoxide. This again proves that sodium methoxide should act as catalyst in the reaction rather than sodium hydroxide.

Table 1. Effect of water content in GL on catalytic performance of sodium methoxide for the transesterification between GL and DMC.

Water content in GL (wt%)

XGL(%)

YGD (%)

YGC (%)

0.0

99

75

22

2.5

91

70

21

5.0

76

60

16

7.5

69

57

12

11.0

61

41

20

Reaction conditions: DMC/GL molar ratio: 2:1; catalyst amount: 3 wt% (to GL weight); reaction temperature: 85 °C; reaction time: 120 min.

Comments 2: Moreover, the authors don’t know how to do arrow pushing for an organic reaction. For example, in Mechanism scheme, in S3, the arrow should start from the glycerol OH not the methoxide. For S4, the arrow should not point the bond but the carbon center. S5 and S6 looks even more weird. If you do arrow pushing, you should do it right. And the authors should not use such gigantic and messy structures in your equations, you can use either ACS style or other style that looks good.

Response: Many many thanks are due to the learned reviewer for the style suggestion and pointing out errors in our reaction scheme. The scheme depicting the mechanism of action has been revised. In the third step (S3), the arrow has now been indicated to start from the glycerol –OH and not the methoxide. In S4, the arrow has now been drawn such that it indicates the carbon centre and not the bond. We have tried our level best to improve the steps, S5 and S6. ACS style has been adopted for the whole mechanism scheme (Scheme 1) and the reaction scheme (Scheme 2).

Comments 3: The transformation is not new at all, as long as you have a base, you should somewhat catalyze this reaction, please check previous publications like, 1, Green Chem 10.1039/c2gc35992h 2, Catal. Sci. Technol., 2013, 3, 3242.

Response: Thanks are due to the learned reviewer for this comment. The trans-esterification of glycerol with dimethyl carbonate is not new (as pointed out by the reviewer) because similar kinds of transformations have been reported previously using different types of bases and basic materials as catalysts. However, this is the first kind of report wherein trans-esterification between glycerol and dimethyl carbonate has been achieved by means of the use of commercially available sodium methoxide as a catalyst. The present conversion using sodium methoxide has same glycerol conversion (99%) with a greater glycidol yield (75%) at lower temperature 85 °C for a smaller reaction time of 2 hrs in comparison to the conversion carried out by Ochoa-Gomez and coworkers [Green Chemistry, DOI: 10.1039/c0xx00000x], wherein the authors have reported same glycerol conversion (99%) with a lower glycidol yield of 6-10% for the conversion carried out at higher temperature (90 °C) for a longer reaction time of 4 hrs. This indicates that sodium methoxide has worked well as a catalyst. Additionally, the present one-pot sodium methoxide catalyzed trans-esterification has shown better glycidol yield and glycerol conversion at a lower reaction temperature than the trans-esterification carried out by using Mg/Zr/Sr mixed oxide base catalysts by Parameswaram et al. (glycerol conversion: 96%, glycidol yield: 40% and reaction temperature: 90 °C) [Catalysis Science and Technology. DOI: 10.1039/C3CY00532A]. These results do indicate that sodium methoxide used in the present study has had a pronounced effect on the catalysis of the trans-esterification of glycerol with dimethyl carbonate.

We now hope that the manuscript is revised appropriately in accordance with the comments of the learned reviewers. Also, explanations to the queries are expected to satisfy the learned reviewer. Therefore, we hope that our manuscript will be accepted for publication in Catalysts this time around.
